# Complex Impacts of Traffic Citations on Road Safety

**Mahdi Rezapour * and Khaled Ksaiabti**

Wyoming Technology Transfer Center, Laramie, WY 82071, USA; khaled@uwyo.edu
* Correspondence: rezapour2088@yahoo.com

**Abstract:** Wyoming has one of the highest fatality rates and lowest enforcement rates in the U.S. Thus, this study was conducted to see if there is any link between various citation types on the equivalent property damage only (EPDO) crashes, while taking into account interaction terms between various crashes and citation characteristics. To achieve the objective, we disaggregated the citations data into their related types, while aggregating the crash dataset across each segment through the Eastern direction of the interstate 80 in Wyoming. The results highlighted that there are important relations between types of crashes and related citations. For instance, it was found that there are (1) significant interaction terms between driving-too-fast-for-condition citations and underride crashes, (2) total number of commercial citations and commercial motor vehicle (CMV) crashes, and (3) between seatbelt citations and seatbelt-related crashes, highlighting, possibly, that the performance of the highway patrol was adjusted based on the specific roadway concerns, or there is indeed a significant link between related citation and crash types. While conducting the model with no interaction term, the results of comparison highlighted that the model erroneously highlight positive associations between citation types and the EPDO crashes. This is one of the earliest studies that considered disaggregated citation data and aggregated them with the crash data to highlight the important relationship between the citations and crashes.

**Keywords:** citation; crash severity; negative binomial reduced rank model; traffic safety; interaction terms

## 1. Introduction

Reducing motor vehicle crashes (MVCs) is a key concern for transportation engineers due to the high number of fatalities. Every year, more than a million people die in MVC crashes around the world, while 50 million more are severely injured on roadways [1]. In the U.S. alone, more than 37,000 people die every year as a result of road crashes [2], while drivers' behavior and actions account for more than 94% of the total causes of crashes [3].

Various approaches have been taken to address errors in drivers' actions and behaviors, which account for the majority of crashes. One of the main approaches is traffic enforcements or citations. The impact of traffic citations on traffic safety is mainly related to the theory of offending, which expresses the idea of crime in terms of costs and benefits of committing that crime [4]. Thus, it is expected that the intention of committing a crime is inversely correlated to the perceived costs. It is reasonable to apply the theory of offending for traffic violations as well. How could traffic citations be used for modeling crash severity or frequency? How could the performance of the highway patrol (HP) be evaluated to improve their efficiency? These are just some of questions that this study seeks to answer.

Despite the importance of traffic enforcement on road safety, unlocking a real relationship between traffic citations and traffic safety is challenging as officers are often impacted by personal preference [5], or citations might be influenced by revenue generation [6]; in addition, it is expected that the enforcement efforts are focused on locations with higher associated risk.

Another challenge of studying the impact of citation data is related to the lack of available information regarding the citation dataset. That is why a large body of the

literature review has focused on a reduction of the crash severity by means of regulators and technological innovations such as seatbelt use instead, see for instance [7], and significantly fewer studies have been conducted on the importance of citation types on roadway safety. Despite all these challenges, it might still be plausible to evaluate the impact of citations on roadway safety if the right methodological approach, such as considering a disaggregated citation dataset, is employed.

Despite the importance of citations on changing drivers' behaviors, most of the limited past studies used the citation data by just considering the total number of citations for their modeling. However, it is also important to understand how specific types of citation might impact particular type of crashes. Is there any link, for instance, between numbers of seatbelt-related citations on the injury severity of crashes due to lack of seatbelt use?

If the highway patrol focused on issuance of commercial motor vehicles (CMVs), does it, in any way, impact the safety of the CMV in particular? Those questions are especially important, as most of past studies being conducted to study the impact of traffic citation on traffic safety were barely involved with the impact of specific types of citation on crashes.

The importance of the use of specific types of citation while modeling traffic crashes could be linked to the study conducted in the literature [8]. Although the idea of that study was implemented for crime, it is expected that it could be applied for our case study too. The idea states that, generally, areas with the highest rate of enforcement are expected to experience the lowest rate of commercial robbery. Furthermore, the impact is linked to the fact that by stopping and questioning suspicious citizens, enforcers are more likely to find a fugitive, or in our case, a person more likely to be involved in a specific type of future crash. Therefore, based on the concept, when a particular group is targeted by highway patrol officers, those risky groups might be less likely to be involved in crashes.

Therefore, in this study, to have a better approach, all crash and citation datasets were considered, and that information was aggregated across each segment of our case study of I-80 E. In addition, the interaction terms were considered across those specific citations and crashes. This is one of the earliest studies that have accessed and used precise citations, in addition to crash datasets, to better understand the impact of various citation types on crash types.

## 2. Literature Review

The following paragraphs present some of the studies conducted using citation data. First, studies regarding the demographic characteristics of drivers, which make them likelier to receive any type of citation, are presented. The second part presents studies that discussed the impact of citations, in relation to crashes.

The characteristics of drivers who received tickets were evaluated [9]. The results highlighted that a number of various ticket types are related to the socio-demographic characteristics of drivers. In another study, gender and age-related differences in attitudes toward traffic laws and traffic violations were evaluated [10]. The results highlighted male and younger drivers show a lower level of motivation to comply with traffic laws compared with their counterparts. Self-report of safe-driving behavior in relationship to age, sex and education was evaluated [11]. The results highlighted that female drivers show a higher rate of seatbelt use, observing speed limits and abstaining from drinking and driving.

The characteristics of speeding drivers, who were traveling 15 mph above the posted speed limit was evaluated in [12]. License plates were used to identify vehicle owners and identify information, regarding the age, gender and vehicle make and model. The speeders were younger, driving with newer vehicles and had more speeding violations, as well as being involved in 60% more crashes.

Regarding the impact of citation on the safety of the roadway, the results of the limited literature highlighted mixed results; while some highlighted positive association, others highlighted negative or no associations. For instance, a dramatic decrease of 35% in Oregon state police officers, due to budget cuts, resulted in a significant decrease in the number of citations [13]. On the other hand, results highlighted that a dramatic decrease in

enforcement is associated with a significant increase in injuries and fatalities. Another study highlighted that while large-scale enforcement reduces crashes, small-scale enforcement has no impact on roadway safety [14].

In another study, to check the impact of the citation on the roadway safety, a sharp increase during the Click-it-or-Ticket (CIOT) program was observed [15]. It was assumed that ticket issuance is exogenous. The results highlighted that traffic tickets significantly reduce crashes and nonfatal injuries.

Aggressive traffic enforcement was used in another study to see if enforcement could reduce injury collision fatalities, especially for those with speed-related injuries [16]. The results highlighted that a significant increase in issued citations is correlated with a reduction in motor vehicle crashes and injury collision and fatalities, especially fatalities due to speeding.

Due to various reasons, especially the fact that most past studies did not have access to comprehensive citations data, or the subjective performance of highway patrol in allocation of resources to the riskier locations, very few studies considered disaggregated citations in crash analysis. Those shortcomings make it hard for policy makers, especially highway patrol, to have a better understanding regarding how to allocate the resources, or even whether their performance has any impact on roadway safety. Thus, this study is conducted to address the aforementioned limitations. In addition, this is one of the first studies to access specific traffic citations. Along with connecting types of citations to the related crash types, we used related interaction terms. For instance, to evaluate the impact of seatbelt citation, the interaction of that citation and crashes involving seatbelt use was considered.

### 3. Data

This study took advantage of variation across different citation and crash types throughout the eastern direction of I-80. Fixed segmentation of a 1-mile segment was used to allocate the datasets to each segment. The crash data came from the WYDOT, while the citation data were collected and obtained from Wyoming highway patrol (WHP) through WYDOT. The two datasets were matched and aggregated across each mile based on the common columns of highway ID and mileposts. To be able to make use of important crash information, such as driver gender involved in a crash, along with the citation data, the average of all characteristics of drivers involved in a crash at a highway segment were aggregated and used.

Each citation provided information regarding the time, date, location and highway system of citations. Initially, there were more than a thousand different types of citations, where they were categorized by an operator and divided into different categories, such as speeding, driving under the influence (DUI), crash predictors, vehicle related, weight related, hours of service (HOS), and red flag (RF), and if a citation did not fall into any category, it was categorized under "others". Three years of data for both crash and citations were used, 2015–2018.

To have a better evaluation of both aspects of crash frequency and severity, crash severity and frequency were aggregated using equivalent property damage (EPDO) as the response. It should be reiterated that, for crash observation, the mean of the observation within a one-mile segment was used, while for citation, the sum of the issued citation was used. That is because the interest is in the mean of the incident, such as underride crashes versus other types of crashes, while for citation, the interest in the sum of the total issued citations. There is an exception of gender for citation data, for which the average was used, and the response of EPDO for crash where the sum was used. For instance, the maximum value of 1 for a segment for crashes mean that either all crashes were experiencing the predictor's extreme values, older drivers, or there was a single value of that observation, see Table 1.

**Table 1.** Descriptive summary of important predictors related to citations and crashes.

| Type | Mean/Sum | Variance | Min | Max |
|---|---|---|---|---|
| *Crash data* | | | | |
| Sum, EPDO crash | 432 | 106,258 | 1 | 1825 |
| Age category, older than 42 years old (vs. others *) | 0.44 | 0.070 | 0 | 1 |
| Belting status of the drivers in crashes, not belted (vs. others *) | 0.03 | 0.009 | 0 | 1 |
| Commercial-vehicles, total across each segment | 78.75 | 6047 | 0 | 737 |
| Hazardous-material vehicles (vs. others *) | 0.31 | 0.062 | 0 | 1 |
| Gender for crashes, female (vs. male *) | 0.20 | 0.037 | 0 | 1 |
| Residency, non-Wyoming residence * (vs. Wyoming residence) | 0.25 | 0.060 | 0 | 1 |
| Too-fast-for-condition, true (vs. false *) | 0.28 | 0.203 | 0 | 1 |
| Vicinity of the drivers, live in the vicinity of 25 miles or less from the crash scene (vs. others *) | 0.10 | 0.0380 | 0 | 1 |
| Underride, true (vs. false *) | 0.13 | 0.04 | 0 | 1 |
| AADT, continuous | 6168 | 1,679,287 | 180 | 12,441 |
| *Citation data* | | | | |
| Sum, DUI (vs. others *) | 2.15 | 14.570 | 0 | 48 |
| Sum, None-peed citation (vs. others *) | 65.54 | 5668 | 0 | 899 |
| Sum, HOS (vs. others *) | 0.74 | 11.526 | 0 | 37 |
| Sum, CP citations (vs. others *) | 5.04 | 35.671 | 0 | 54 |
| Sum, Gender for citation, female (vs. male *) | 0.27 | 0.005 | 0 | 0.55 |
| Sum, Speed-related citations (vs. others *) | 83.83 | 14,488 | 1 | 1424 |
| Residency, non-Wyoming residence (vs. Wyoming residence) | 0.33 | 0.220 | 0 | 1 |
| Sum, Other's citations (vs. others *) | 16.58 | 664.133 | 0 | 372 |
| Sum, Too-fast-for-condition (vs. others *) | 0.97 | 2.320 | 0 | 15 |
| Sum, Commercial vehicle (vs. others *) | 76.75 | 6047 | 0 | 737 |
| Sum, Seatbelt related (vs. others *) | 7.56 | 106.432 | 0 | 132 |

\* Reference as 0, No of observations = 402.

Initially, the average of age, 42.5 years was used, which divides the crash data into almost two equal groups, and then that was aggregated across the segments for the binary age indicator for those drivers that were involved in crashes.

A few points should be highlighted regrading traffic citations. Crash predictors' (CP) citations are those CMV citations related to unsafe driving practices that might result in crashes [17]. Those include citations related to reckless driving, improper lane change, improper turn, and failure to obey traffic devices.

Hours of service (HOS) violations, on the other hand, are related to the time when drivers operate the vehicles for more than 14 consecutive hours or violated off duty for 10 consecutive hours, where they receive HOS citations. Other violations or "others" are those citations that did not belong to any category. To differentiate whether we considered the sum or average of variables, we used "sum" for those variables where the total number was used.

## 4. Method

As described by [18], the advantage of the reduced-rank regression (RRR) could be summarized as the use of a combination of explanatory variables, which could provide more explanatory power and at the same time provide a low-dimensional data perspective. Although the RRR has usually been applied to high-dimensional problems, the literature highlighted that it could also be used for low-dimensional models [19].

The explanatory variables are written as $x = (x_1, x_2)$. Therefore, linear equation based on the transformed response of $\eta_i$ is written as:

$$\eta_i = B_1^{\mathsf{T}} x_{1i} + AC^{\mathsf{T}} x_{2i} = B_1^{\mathsf{T}} x_{1i} + A\nu_i \qquad (1)$$

where $x_1$ is a vector of an explanatory variable, which is set as 1 for an intercept for most cases, and $x_2$ is a vector for parameters to be estimated. $x$ and $B$ are partitioned to $\left(x_1^\mathsf{T}, x_2^\mathsf{T}\right)^\mathsf{T}$, and $B = \left(B_1^\mathsf{T} B_2^\mathsf{T}\right)^\mathsf{T}$, respectively. $B_2$ is reduced by replacing that by the reduced-rank regression, while $B_1^\mathsf{T}$ is kept as it is, e.g., for intercept.

On the other hand, $x_2$ is replaced by a smaller matrix of $v_i$, which is based on Equation (1). The process could be summarized as creating matrices of $B_1^\mathsf{T}$, $A$ *and* $C^\mathsf{T}$, updating the matrices and refitting the model where $C^\mathsf{T}$ *and* $x_{2i}$ are used for creating the latent variable of $v_i$, see Equation (1). Now the matrix of $A$ based on corner constraint can be written as:

$$A = \left( \begin{array}{c} I_R \\ \tilde{A} \end{array} \right) \tag{2}$$

On the other hand, $z$ or the working response, is regressed based on the transformed explanatory variables for parameter estimates as:

$$z^{ite} = X_{vlm_{iter}} B \tag{3}$$

where *iter* in $z^{\text{iter}}$ and $X_{vlm_{\text{iter}}}$ are related to parameters at iteration number of "iter". Then using the coefficient related to $x_{2i}$, matrix of $C$ is updated as $C = A^{-1} \times est.coef_{x_{2i}}$, see Equation (1). Therefore, at each iteration after obtaining an updated version of coefficients, the matrix of $C$ is updated.

It should be noted that the $I_R$ or *latvar.mat*, in the Results section and Equation (2), corresponds to the elements of $A$. Based on Equation (1), the latent variable $v$ is built by multiplication of the considered attributes by the constrained coefficients of $C$ as:

$$v = C^T x_2 \tag{4}$$

In the last stage of the iteration process z is made as:

$$z^{(n)} = \eta_i^{(n)} + \left(W_i\right)^{(n)^{-1}} u_i^{(n)} \tag{5}$$

where $(u_i)_j = \frac{\partial \ell_i}{\partial \eta_j}$, is the score vector for *jth* element, and $(W_i)_{jk} = \frac{-\partial^2 \ell_i}{\partial \eta_j \partial \eta_K}$. $(W_i)_{jk}$ measures the amount of information each observation carries, rather than the sum of the information. $n$ is also the number of iterations.

The model's parameter estimations can be summarized as follows [18]:

- Initial values of various negative binomial (NB) parameters, including the distribution's means and dispersion incorporated in $\eta$, are given by grid search. It should be highlighted that means and dispersion of the NB are saved in $\eta$ columns: throughout the process, and the parameters of NB could be used for obtaining the $\eta$ and vice versa.
- Transform the model from generalized least square (GLS) to the ordinary least square (OLS) by multiplying both sides of Equation (5) by the Cholesky decomposition of $U$, to standardize the error terms and remove the correlation.
- After the transformation the model is solved by the OLS.
- For transformation of the explanatory variable, Equation (3), Kronecker matrix using the original explanatory variables and constraints is used to build $X_{vlm}$.
- As the interest, in addition to the explanatory variables' parameters, is obtaining the parameters of the latent variable, two matrices of LV and the original of matrix are used by means of two-phase regression. First, the LV is conducted, and the matrix of $C$ is used for the next modeling of the original model. In this way, the estimated parameters of the attributes are kept, and thus have both estimates of the parameters and latent variable.
- In each iteration the updated version of matrices of $C$ and $A$ will be used and replace the older version, see Equation (1).

- Residual is estimated by the difference between the *z*, Equation (5), and the fitted value based on the parameters.
- Standard Deviation (SD) of the parameters is obtained by diagonal matrix of the inverse of the *R* matrix of the QR matrix decomposition, for estimation of *p*-value. The QR will be obtained from the output of the OLS method.
- The convergence would be achieved when in the iteration process the variation across two consecutive log likelihoods, or changes in parameter estimates, are less than the predefined minimum value of $\varepsilon$.
- The process of the RR-VGLM is based on iteratively reweighted least square (IRLS).

## 5. Results

Two models were considered, including the model considering the interaction terms and model with only main effects to highlight the differences in terms of estimated parameters, see Table 2. All variables were considered in the latent variable term due to the fact that all are related to interaction terms from citations. As a result, only intercept was assigned to the $x_{1i}$ matrix, see Equation (1).

**Table 2.** Estimated parameters of the RRR, considering interaction terms and no interaction terms.

| Models. | | A | | | B | | |
|---|---|---|---|---|---|---|---|
| Coefficients | | **Interaction Terms Are Considered** | | | **Interaction Is Not Considered** | | |
| | | **Estimate** | **Std. Error** | ***p*-Value** | **Estimate** | **Std. Error** | ***p*-Value** |
| **Main effects** | | | | | | | |
| Crash | *latvar.mat* or *A matrix* | 0.14 | 0.011 | <0.005 | 0.04 | 0.011 | <0.005 |
| | Driver age | 0.82 | 0.205 | <0.005 | 0.91 | 0.173 | <0.005 |
| | Belting status of drivers | 0.75 | 0.476 | 0.006 | 0.34 | 0.507 | 0.25 |
| | Commercial vehicles | 1.30 | 0.877 | 0.07 | 0.53 | 1.139 | 0.34 |
| | Hazardous-material vehicles | 0.43 | 0.871 | 0.3 | 1.16 | 1.137 | 0.15 |
| | Gender | 5.52 | 0.721 | <0.005 | 1.47 | 0.243 | <0.005 |
| | Residency | 0.04 | 0.180 | 0.4 | 0.16 | 0.208 | 0.22 |
| | Too-fast-for-condition | 0.36 | 0.117 | <0.005 | −0.14 | 0.101 | 0.08 |
| | Underride | 0.23 | 0.202 | 0.1 | 0.51 | 0.237 | <0.005 |
| | AADT | 0.001 | 0.00003 | <0.005 | 0.001 | 0.000 | <0.005 |
| Citation | DUI | 0.08 | 0.032 | <0.005 | 0.04 | 0.020 | 0.0201 |
| | None-peed | 0.02 | 0.003 | <0.005 | 0.003 | 0.001 | 0.02 |
| | HOS | 0.03 | 0.016 | 0.04 | 0.03 | 0.017 | 0.02 |
| | CP citations | 0.05 | 0.012 | <0.005 | 0.02 | 0.009 | 0.04 |
| | Gender | 4.61 | 0.630 | <0.005 | 4.24 | 0.568 | <0.005 |
| | Speed-related | 0.002 | 0.001 | 0.06 | −0.002 | 0.001 | 0.02 |
| | Residency | 0.06 | 0.102 | 0.3 | 0.13 | 0.095 | 0.08 |
| | Other's | 0.003 | 0.004 | 0.3 | −0.01 | 0.005 | <0.005 |
| | Too fast for condition | 0.21 | 0.027 | <0.005 | 0.27 | 0.029 | <0.005 |
| | Commercial vehicle | 0.002 | 0.001 | 0.02 | 0.001 | 0.001 | 0.13 |
| | Seatbelt related | 0.01 | 0.006 | 0.06 | 0.01 | 0.008 | 0.15 |
| **Interaction terms** | | | | | | | |
| Drivers' demographic characteristics | DUI citation ×Driver age, crash | −0.14 | 0.062 | 0.1 | —— | —— | —— |
| | Residency citation × Residency crash | −0.49 | 0.318 | 0.06 | | | |
| | Gender, citation × Gender, crash | −16.69 | 2.434 | <0.005 | —— | —— | —— |

**Table 2.** *Cont.*

| Models. | | A | | | B | | |
|---|---|---|---|---|---|---|---|
| **Coefficients** | | **Interaction Terms Are Considered** | | | **Interaction Is Not Considered** | | |
| | | **Estimate** | **Std. Error** | **$p$-Value** | **Estimate** | **Std. Error** | **$p$-Value** |
| CMV citations | HOS citation × Commercial-vehicles crashes | −0.22 | 0.063 | <0.005 | —— | —— | —— |
| | CP citations × Hazardous material | −0.07 | 0.027 | <0.005 | —— | —— | —— |
| | Commercial vehicle citations × Commercial-vehicles crashes | −0.01 | 0.002 | <0.005 | —— | —— | —— |
| Speed-related actions | Speed-related citations× Driver age, in crashes | −0.01 | 0.002 | <0.005 | —— | —— | —— |
| | Too-fast-for-condition citations × Underride crash | −0.25 | 0.182 | 0.09 | —— | —— | —— |
| Others | Other's citations × Too-fast-for-condition crashes | −0.02 | 0.006 | <0.005 | —— | —— | —— |
| | Seatbelt related citations × Belting status of the drivers in crashes | −0.15 | 0.031 | <0.005 | —— | —— | —— |
| | AADT × Non speed related citations | $-3 \times 10^{-6}$ | $4 \times 10^{-7}$ | <0.005 | —— | —— | —— |
| Log-likelihood: −2710 on 757 degrees of freedom, AIC = 5485 | | | | | −2829 on 768 degrees of freedom, AIC = 5701 | | |

For interpretation of the interaction term between two variables, the main effects and interaction term itself should be considered. Therefore, for interaction of $A \times B$, for instance, it is clear that all parameters of "$A$" + "$B$" + "$A \times B$" are considered. A total of 11 interaction terms between citations and crash characteristics was considered, being found to be important. For ease of parameter discussion, the estimates are presented in four main subcategories and the discussion is also given based on those subcategories. All interpretations are based on the fact that variation across the majority of interaction estimates, and main effect of the citation types, is so large that, almost always, the negative estimate of interaction terms $A \times B$, will cover the positive sign of the citation main effects, unless otherwise stated.

### 5.1. Drivers' Demographic Characteristics

In this section, three demographic characteristics of drivers in crashes, including driver age, gender and residency, for both citation and crash data, were considered.

The results of interaction terms between DUI citations and driver age in crashes highlights that for older age groups, when the age is kept constant at a specific value, an increase in DUI citations decreases the EPDO crashes.

The results might seem intuitive, especially for younger drivers, as they are likelier to drink and drive. It should be recalled that the original dataset had a binary predictor for age and in the aggregated dataset being used in this study, the majority of age observations had non-zero values (older drivers). For instance, if the segment of 0–1 mile included two older drivers, as 1, and younger, as 0, involved in a crash, the mean of 0.5 is used in that segment.

The interaction between gender, and residency of drivers involved in citations and crashes both highlight the importance of targeting the right group for crash reduction. Although it is not practical to target drivers with unique characteristics, e.g., Wyoming-plate drivers, the results highlighted that an increase in citation for older gender or Wyoming residency results in a reduction in the EPDO. That might be related to the endogeneity of WHP performance based on the crash conditions.

In other words, there is a link between the gender and residency of drivers involved in crashes and committing violations, so considering that, the interaction terms could result in a reduction in the EPDO.

### 5.2. CMV Citations

Moving to the CMV interaction terms, the results of three interaction terms highlighted that both citations and crashes correspond to the same subject of CMV. The results highlighted that an increase in citations related to HOS, which is related to CMV, along with an increase in the CMV crashes, result in a decrease in the EPDO crashes.

Although the result might highlight the endogeneity of the performance of the highway patrol for issuing more CMV-related citations, in case of an increase in the CMV crashes, still, if their performance is aligned with the situation on the interstate, it is expected that the severity of crashes will be reduced. On the other hand, if the CP citations increase at the locations with high hazardous-material-vehicles crashes, it is expected that the EPDO will be reduced.

### 5.3. Speeding-Related Actions

Speeding has been recognized as a major traffic-safety problem [20]. The first interaction is related to driver age and speed-related citations. The results highlighted that if age is kept fixed at a certain value and speeding-related citations increase, the EPDO crashes are expected to decrease, $\hat{\beta}_{speed-related\ citation \times driver\ age} = -0.01$ versus $\hat{\beta}_{speed-related\ citation} = 0.0020$. Based on the point estimates, the reduction impact of speeding-related citations is much greater if the citation is incurred by the younger driver, $\hat{\beta}_{age} = 0.82$.

It was also found that there is a significant interaction in terms of driving too fast for conditions and underride crashes. Underride crashes occur when passenger cars go under other vehicles, mainly semi-trucks. Various factors could be linked to underride crashes, including brake failure, drivers' lack of attention, or driving too fast for conditions. All these factors could result in drivers' lack of control of vehicles and consequently results in underride crashes. Our results highlighted the effectiveness of driving-too-fast-for-conditions citations in reducing EPDO crashes due to underride crashes ($\hat{\beta}_{interaction\ terms} \times x_{underride\ crashes \times too fast\ for\ conditions} + \hat{\beta}_{underride\ crashes} \times x_{underride\ crashes} + \hat{\beta}_{too fast\ for\ conditions} \times x_{too fast\ for\ conditions}$), where $\hat{\beta}$ stands for underride crashes, too fast for conditions and interaction terms are 0.23, 0.21, and $-0.25$, respectively.

### 5.4. Others

Regarding the last category in Table 2 or others, other types of citation were found to reduce EPDO crashes, while interacting with driving-too-fast-for-condition crashes, $-0.02$ versus 0.003 for interaction terms and main effect of other's citations, respectively. Therefore, while keeping the crash characteristics of driving-too-fast-for-conditions constant, an increase in the number of others citations results in a reduction in EPDO crashes, $\hat{\beta}_{others\ citation} = 0.003$ versus $\hat{\beta}_{others\ citation \times Too-fast for condition\ crash} = 0.003$.

Moving to the interaction terms between seatbelt citation and seatbelt crashes, which could not be fit in any category. The results highlighted that issuance of seatbelt citations reduces EPDO crashes, while interacting with drivers being not buckled in crashes: $-0.15$ for interaction terms versus 0.01 for the seatbelt related citation. Again, this impact might be due to the adjustment in WHP performance in targeting groups with higher crashes. In other words, when the number of seatbelt crashes goes up in a segment, the number of related citations also increases in the same segment.

The last interaction term is related to the interaction between traffic and non-speed-related citations. The result highlighted that for segments with higher average annual daily traffic (AADT) and higher non-speed-related citations, it is expected to have segments with higher EPDO crashes. That is likely to be related to the endogeneity of the highway patrol, influencing their performance based on traffic. In other words, at segments with higher AADT, it is likelier to have higher EPDO crashes, while WHP also issued more non-speed-related citations.

In terms of the impact of the latent variable, the impact is complex to interpret as it accounts for a variety of predictors. However, again, the positive signs might be considered as a general positive impact of the citation latent variable on the segment EPDO crashes due to the endogeneity.

For a comparison purpose, the model with no interaction terms was implemented to see the differences, especially in terms of point estimates. As could be seen from Table 2, not considering the interaction terms in model B, out of 11 considered citation effects, the impact of only "others", and speed-related citations, are negative. As a result, not considering the interaction terms, the model considers almost all citation types to be positively correlated with the EPDO crashes, which would, consequently, result in a lack of understanding and misguided results.

Finally, to provide a better vision regarding the interactive relationships between various predictors, Figure 1 is provided to link citation with related crash types. For instance, consider seatbelt citation, which interacts with seatbelt status at the time of crashes, or CMV citation, which interacts with CMV crashes.

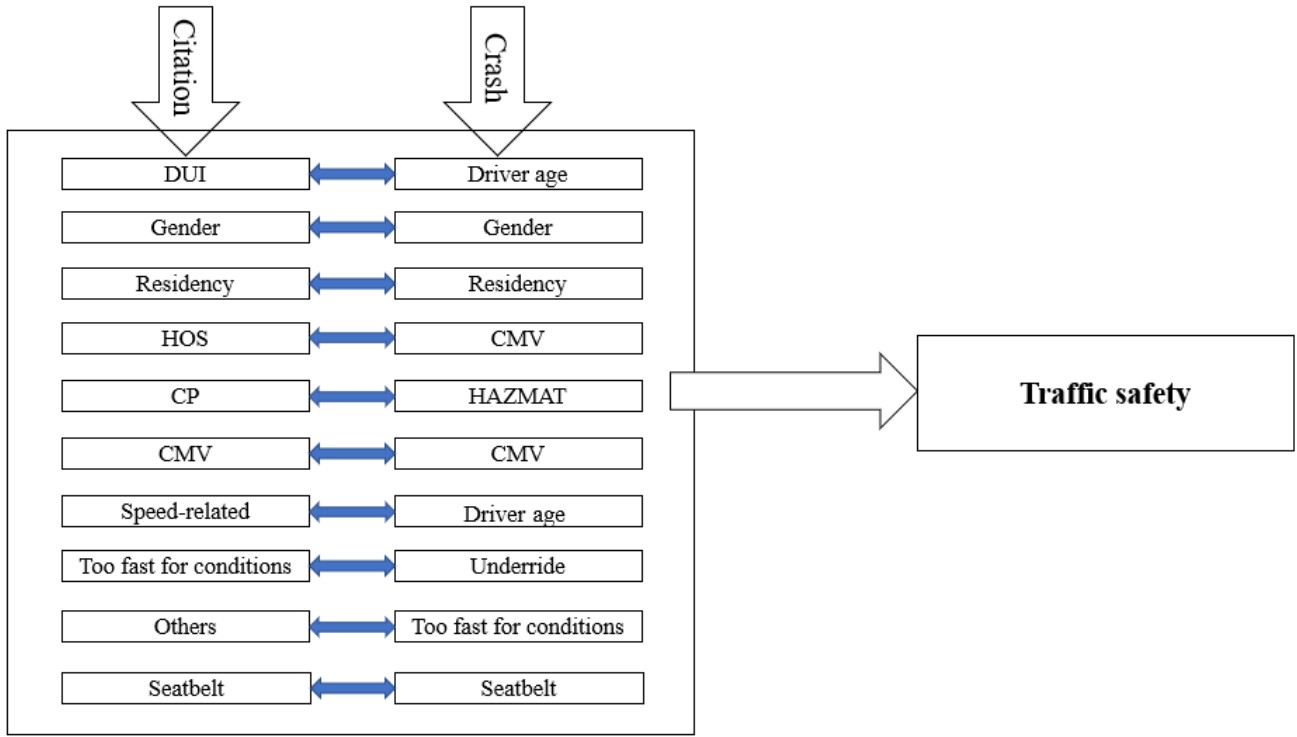

**Figure 1.** Important interaction terms across citations and crash characteristics.

## 6. Discussion

Reducing severe crashes is a major concern in Wyoming due to the high fatality rate. The high fatality rate has been assigned to various factors, such as mountainous topography of the state and high truck-through traffic. Extensive efforts have been made to improve the road safety, especially in terms of reduction in severe crashes. Special attention has been paid to traffic enforcement due to the main aspect of changing drivers' behavior and

actions, which are the main factors in crashes. The adjustment of behaviors is believed to contribute significantly to the higher safety of the highway system in the state.

Thus, this study was conducted to evaluate the effectiveness of various enforcement types in reduction of the EPDO crashes. To have a better vision about the state of the crash magnitude, the crash frequency and crash severity were aggregated for creating the EPDO measure. The EPDO is known to be over-dispersed, which makes methods such as Poisson unsuitable for the modeling due to the sparse dataset nature.

The negative binomial, in the context of the reduced-rank regression (RRR) model, was employed. The IRLS was used to estimate both the parameters of the latent variable, latvar.mat, or *A* matrix, and also to estimate the parameters inside that latent variable. The reason for the use of the latent variable was that the impact of citations on the severity of crashes might not be measured directly, so it might be an unobserved or latent trait. The latent trait depends on various interaction terms and main effects of enforcement and other explanatory variables. Therefore, the latent variable of enforcement is a combination of those predictors.

The number of studies that considered citation data for the analysis of factors contributing to the severity of crashes is limited. Especially, the number of studies that considered specific types of citation, while modeling the severity or frequency of crashes, is limited. The studies that evaluated the effect of citations found inconsistent results: while some highlighted negative correlation between traffic safety and number of issued citations, others find no impact or positive correlation. In summary, the shortcomings of the previous studies could be summarized as: (1) solely the use of self-reported data, e.g., survey, (2) non-consideration of crash while implementing the citation data, (3) non-consideration of the citation data in terms of types of citations.

As a result, it is unclear in the literature whether the traffic citations fulfill the intended objectives of an improvement in traffic safety or reduction in the specific types of crashes. The impact might be due to the endogeneity of the performance of the highway patrol in allocation of resources to specific parts of road segments, due to predominance of particular drivers' characteristics at those locations, e.g., DUI or speeding.

The results of this study highlighted that while non-speed citations interact with traffic, the higher number of that citation is associated with higher EPDO crashes. On the other hand, the specific types of citations have a preventive effect on EPDO crashes, while interacting with other crash characteristics. For instance, it was found that HOS, CP and total number of CMV citations, while interacting with various CMV characteristics, reduce EPDO crashes. Further, seatbelt citations interact with the seatbelt status of drivers in crashes, or DUI citations interact with age characteristics while influencing EPDO crashes.

It is expected that committing a traffic violation is a matter of gain–loss consideration before making any action and decision. Therefore, by increasing the presence of highway patrol on roadways, at least the perception regarding the cost of violating traffic laws would be increased, which consequently could decrease the likelihood and severity of those crashes. The results also highlighted that targeting specific violations, while interacting with the related crash types, could reduce EPDO crashes. That is especially important, as it was highlighted in the literature that traffic violations to be correlated with delinquent behaviors [21], and also traffic violation, could be used as a means to measure the likelihood of future crashes [22–24].

In summary, although it is expected that an increase in citations is an artifact of an increase in traffic or crashes, the results showed that, while the endogeneity exists, subjectivity of the highway patrol in targeting specific individual types of violators still has a preventive effect on different types of crashes.

The results highlighted that highway patrol officers could contribute to the roadway safety by not only increasing the associated costs against associated so-called benefits, but also targeting specific violation due to a possible increase in the related crashes; for instance, an increase in the number of seatbelt citations, in case of an increase in those crashes. More studies are encouraged to focus more on specific types of citation and necessarily

consider the interaction terms between various types of citations and particular types of related crashes.

*6.1. Recommendations*

WYDOT regularly provide WHP with the necessary crash information, so WHP could conduct their responsibilities in the most efficient way. As it was found that WHP performs well in targeting specific types of violators to increase the safety of roadways, it is important to provide important crash data trends in a frequent manner and also to include the important information for them to reduce the crash severity and frequency. Some of the recommendations for both groups could be outlined as follows.

6.1.1. WYDOT

It was found that it is especially important for WHP to be up to date regarding the trend and particular types of crashes, so WHP could target the violators of those specific types of traffic crashes, e.g., an increase in CMV citations, in case of an increase in CMV crashes.

Further, if the non-seatbelt-use trend increases in crashes during a particular timeframe, the report of WYDOT informs WHP about the changes, so WHP could allocate their resources more appropriately. The data provided to WHP, especially, should include the trend, in terms of crash frequency and severity for CMV, seatbelt, various driver actions, DUI crashes and crashes based on different age groups.

6.1.2. WHP

The report provided by WYDOT increases the awareness regarding the trend of various crash types, so WHP could deliver effective enforcement countermeasures by targeting risky groups. It is important for WHP to link the crash data with the summary of the issued citations, so the citations will be adjusted based on crashes. Previously, studies highlighted that violators with specific violations are possibly more likely to be involved in crashes due to those violations. Therefore, targeting specific violators is a rational way to target risky drivers in future crashes.

In case of an increase in the trend of a particular crash, the enforcement related to that crash should be increased, and possible incentives might be given to the officers for issuing those particular citations. For instance, in the case of observing an increase in CMV crashes, citations, such as CP or CMV citations, should be increased.

**Author Contributions:** Methodology, M.R.; supervision, K.K. All authors have read and agreed to the published version of the manuscript.

**Funding:** This research was funded by WYDOT.

**Institutional Review Board Statement:** Not applicable.

**Informed Consent Statement:** Not applicable.

**Data Availability Statement:** Not applicable.

**Conflicts of Interest:** The authors declare no conflict of interest.

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
