# Peer review of "Complex Impacts of Traffic Citations on Road Safety"

_futuretransp, doi:10.3390/futuretransp2020022_

Round 1

Reviewer 1 Report

This paper well presents the analysis of the impacts of traffic citations on traffic crashes. Though it is well organized, I have a few comments that the authors need to address for improved paper quality:

1) The title said the severity of crashes, but the whole analysis in the paper only covers the equivalent 8 property damage only (EPDO) crashes. I would suggest the authors make changes to the title to reflect the main topics of this study;

2) L. 146, mean of observations, mean of the incident (L.148), I think the authors want to indicate the number of crashes, please clarify those terms to be able to be treated as the response variable in NB model;

3) L.251, younger age group, it is a vague terms, please use age categories to describe this age group;

4) Define the dashed dots in Table 2;

5) incomplete Figure 1, severity of crashes? I do not see the severity crashes; What does each arrow mean in this figure, I do not quite get it.

Author Response

This paper well presents the analysis of the impacts of traffic citations on traffic crashes. Though it is well organized, I have a few comments that the authors need to address for improved paper quality:

  • The title said the severity of crashes, but the whole analysis in the paper only covers the equivalent property damage only (EPDO) crashes. I would suggest the authors make changes to the title to reflect the main topics of this study;

We agree and we changed the title to address your concern as “

Impacts of traffic citations on the road safety”

  • 146, mean of observations, mean of the incident (L.148), I think the authors want to indicate the number of crashes, please clarify those terms to be able to be treated as the response variable in NB model;

We agree with the reviewer. We added the point of

To clarify Table 1, in case of using the total of observations in the data, the term “sum” was used to highlight that. “

Also we added “sum” to variables to be clearer. That is because we have both sum and mean.

  • 251, younger age group, it is a vague terms, please use age categories to describe this age group;

The category is changed to “Age category, older than the 42 years old (vs others*)”

  • Define the dashed dots in Table 2;

Actually, those are for the model that interaction terms were not considered. So, for that model there is no interaction terms so that is why there are dashes.

5) incomplete Figure 1, severity of crashes? I do not see the severity crashes; What does each arrow mean in this figure, I do not quite get it.

The figure has been amended. The point is explained by “It should be noted that the related citation is linked to the related crash types. For instance, consider seatbelt citation which is interacting with seatbelt status at the time of crashes, or CMV citation which is interacting with CMV crashes. “

We appreciate the reviewer’s time and please let us know if you have any further concern.

Reviewer 2 Report

The paper needs some modification:

  1. The problem statement should be written more clearly.
  2. What is the practical contribution of the study conducted?
  3. The contribution of the paper seems to be too low. 

Author Response

  1. The problem statement should be written more clearly.
  2. What is the practical contribution of the study conducted?
  3. The contribution of the paper seems to be too low.

The point is added as follows

Thus, this study is conducted to address the aforementioned limitations by focusing on specific points as follows: this is one of the first studies access specific traffic citations. In addition to connect the types of citations to the related crash types, we used interaction terms.  For instance, to evaluate the impact of seatbelt citation, the interaction of that citation and crashes involved seatbelt use was considered.

It should be noted that this is one of the earliest studies that could achieve the connection between specific types of citations and related crashes!

Thanks for your time and please let us know if you have any further concerns.

Reviewer 3 Report

The authors present two sets of data (accidents and citations) referring to homogeneous stretches of road and referring to the same time interval (3 years); however, it would have been more appropriate to use different reference time intervals for the data, given that generally the penalties due to citations have effects that are deferred over time.

It is not clear how the authors aggregate the characteristics of the drivers along the segments to use crash information.

Evaluating the effects of citations on road safety is a very interesting topic. The authors present two sets of data (accidents and citations) referring to homogeneous stretches of road and referring to the same time interval (3 years); however, it would have been more appropriate to use different reference time intervals for the data, given that generally the penalties due to citations have different effects over time.

The Method section is very confused; please, provide a more detailed explanation of the methodological aspects, commenting appropriately on the equations adopted and the variables of the models.

I suggest reviewing the paper and investigating the methodological aspects, especially concerning the inputs of the models and their parameters.

I also suggest considering a “Literature Review” section in which the authors could expand the references.

Author Response

The authors present two sets of data (accidents and citations) referring to homogeneous stretches of road and referring to the same time interval (3 years); however, it would have been more appropriate to use different reference time intervals for the data, given that generally the penalties due to citations have effects that are deferred over time.

That is a very unique point. However, unfortunately we did not have access to the time here. But we will consider that in our future works.

It is not clear how the authors aggregate the characteristics of the drivers along the segments to use crash information.

For instance, consider the drivers gender. If we had two drivers, one male and another female, the mean of those two drivers’ gender would be set at location as 0.5, average of 0 and 1.

Evaluating the effects of citations on road safety is a very interesting topic. The authors present two sets of data (accidents and citations) referring to homogeneous stretches of road and referring to the same time interval (3 years); however, it would have been more appropriate to use different reference time intervals for the data, given that generally the penalties due to citations have different effects over time.

As we elaborated in the previous comment, this is a great point but unfortunately we did not have access to time period and also Wyoming, the case study, has a very low traffic so it would be impractical to disaggregate those observations across various time,

The Method section is very confused; please, provide a more detailed explanation of the methodological aspects, commenting appropriately on the equations adopted and the variables of the models.

We have revisited and made necessary corrections. Please let us know if you have any specific comment,

I suggest reviewing the paper and investigating the methodological aspects, especially concerning the inputs of the models and their parameters.

We agree and as mentioned in the previous response we revisited the method section.

I also suggest considering a “Literature Review” section in which the authors could expand the references.

We agree and have gone over the literature to make sure all is good. It should be noted that this is a first study implemented the methodological approach of using disaggregated citation data so there were not many studies to use but please let us know if you would like us to elaborate on any specific point.

Round 2

Reviewer 2 Report

The scientific merit of the study is too low. Hence, the study should not be accepted at this stage.

Author Response

We are able to amend any portion of the manuscript, if you are able to provide use with some recommendation.